# Salus Platform: A Digital Health Solution Tool for Managing Syphilis Cases in Brazil—A Comparative Analysis

**DOI:** 10.3390/ijerph20075258

**Published:** 2023-03-24

**Authors:** Talita Brito, Thaísa Lima, Aliete Cunha-Oliveira, André Noronha, Cintia Brito, Fernando Farias, Sedir Morais, Jailton Paiva, Cintia Honorato, Paulo Queirós, Sagrario Gómez-Cantarino, Márcia Lucena, Ricardo Valentim

**Affiliations:** 1Laboratory of Technological Innovation in Health, Federal University of Rio Grande do Norte, Natal 59072-970, Brazil; 2Health Sciences Research Unit: Nursing (UICISA: E), Coimbra Nursing School (ESEnfC), 3004-011 Coimbra, Portugal; 3Ministry of Health, Esplanada dos Ministérios, Block G, Headquarters Building, Brasília 70058-900, Brazil; 4Center for Interdisciplinary Studies of the 20th Century (CEIS-20), University of Coimbra, 3000-456 Coimbra, Portugal; 5Department of Pharmacy, Federal University of Rio Grande do Norte, Natal 59072-970, Brazil; 6Department of Physical Education, University of Pernambuco, Recife 52012-570, Brazil; 7Federal Institute of Sergipe, Aracaju 49680-000, Brazil; 8Federal Institute of Rio Grande do Norte, Natal 59015-000, Brazil; 9Hospital of State Servers, Rio de Janeiro 20221-161, Brazil; 10Faculty of Physiotherapy and Nursing, Campus Toledo, University of Castilla-La Mancha, 45071 Toledo, Spain; 11Department of Informatics and Applied Mathematics (DIMAP), Federal University of Rio Grande do Norte, Natal 59078-900, Brazil; 12Department of Biomedical Engineering, Federal University of Rio Grande do Norte, Natal 59077-080, Brazil

**Keywords:** Salus, acquired syphilis, maternal syphilis, congenital syphilis, information system, primary attention, sexual infection, epidemiological monitoring

## Abstract

(1) Introduction: Syphilis is a sexually transmitted infection (STI) that constitutes a serious public health problem in Brazil and worldwide; (2) Methods: This was a descriptive and exploratory study that sought to analyze and compare the characteristics of Brazilian health systems with a new platform (Salus) developed by the Laboratory of Technological Innovation in Health in the scope of notification and management of disease data, including syphilis. In addition, this analysis aimed to assess whether Salus fully meets the necessary data management fields and can be indicated as a tool to improve health management in the context of syphilis in Brazil. (3) Results: In this study, the Salus functionalities were demonstrated and compared with the current Brazilian systems by discovering the existing gaps in the evaluated systems. The gaps found may explain the delay in meeting demands, the difficulty of making routine therapeutic follow-ups, in addition to interference with the vital purpose of follow-up in the epidemiological surveillance of diseases. As a result, Salus demonstrates functionalities that surpass all others and meet case management demands in a superior way to the systems currently used in the country. (4) Conclusions: The Brazilian health information systems related to the response to syphilis do not fulfill the purpose for which they were developed. Instead, they contribute to the fragmentation of health data and information, delays in diagnosis, incomplete case management, and loss of data due to inconsistencies and inadequate reporting. In addition, they are systems without interconnection, which do not articulate epidemiological surveillance actions with primary health care. All these factors may be obscuring accurate data on syphilis in Brazil, resulting in high and unnecessary public spending and late care for users of the Unified Health System (SUS).

## 1. Introduction

### 1.1. Syphilis

Syphilis is a sexually transmitted infection (STI) that is systemic and caused by Treponema pallidum. It constitutes to present a serious public health problem in Brazil and the world [1,2,3,4], despite the existence of methods of prevention, diagnosis, and treatment with an antibiotic, which is widely used as the gold standard for treating syphilis [5,6,7,8,9]. Moreover, it presents substantial morbidity, mortality, and chronic evolution and is a gateway to other STIs [10,11,12,13,14].

The Brazilian notification process of syphilis is still flawed, as it generates fragmented data and contains unjustifiable delays [15,16,17,18,19,20,21]. For example, SINAN (Information System for Notifiable Diseases) data on the consolidation of syphilis are released only once a year, always using data from the previous year. This makes decision-making much more complex regarding the conduct of public health policy to respond to the problem of syphilis in the country [16,22,23,24]. In addition, with a year’s delay in obtaining consolidated information on syphilis, the public health manager will have to make decisions looking to the past. This aggravates public health problems and the population in general [25,26]. According to Ordinance 204 from 17 February 2016, syphilis requires mandatory weekly notification. In addition, according to the Unified Health System (SUS) protocols, a data-sharing flow must be followed in disease management at SUS. However, this deadline is not met, which causes a delay in beginning treatment. However, when it comes to a pregnant woman, the situation worsens. The delay in the intervention and at the beginning of the treatment can cause several damages, such as maternal mortality or vertical transmission of syphilis [27,28].

Thus, in 2017, the Ministry of Health (MoH) invited municipal and state health managers to adhere to the national strategy “Rapid Response to Confronting Syphilis in Care Networks”. This strategy aims to reduce the fragmentation of processes and techniques that guide coping with grievances and developing the care competence and managerial response of the services [14,29]. This strategy became known nationally and internationally as the Syphilis No! Project [3].

In the context of the high demand for disease management systems in Brazil, opportunities have emerged for implementing innovative strategies and devices that focus on the management process and care technologies to intensify the control, fight, and monitoring of syphilis and other sexually transmitted infections transmissible in the country. For this, measures are proposed that optimize time and public money, focusing mainly on the patient’s well-being. Intelligent notification platforms are needed, with management tools that have interoperability between government systems at all levels of care, agility in sending data, and control in a clear, fast, and transparent manner [30]. This is one of the biggest challenges for the control of syphilis cases: the insertion and implementation of tools that articulate the three levels of health care with epidemiological surveillance, guaranteeing easy access to diagnosis, treatment, and follow-up of diseases [12,31,32].

Given these factors, and as a prerogative of the Syphilis No! Project, researchers developed a digital health solution, the Salus Platform, to carry out all case management and clinical follow-up of people with syphilis, including pregnant women, sexual partners, and children who have been exposed, until the case’s conclusion, aiming to end the cycle of transmission of the disease.

Thus, the objective of this study was to analyze the Salus Platform as a digital health solution for the management of syphilis cases, comparing its functionalities with those of some health information systems available in Brazil regarding the response to syphilis. This study is part of the project called “Applied research for intelligent integration aimed at strengthening health networks for a quick response to syphilis” (Syphilis No!), whose main objective is to reduce acquired syphilis, MS, and eliminate cases of CS in Brazil through educational measures of surveillance, health, scientific research, and public policies [33,34].

### 1.2. Salus (Intelligent Disease Monitoring System in Primary Care and Epidemiological Surveillance)

In order to promote health within the population and analyze the problem of managing cases of syphilis in Brazil, the Salus Platform was created by the LAIS/UFRN research group, who developed a technological innovation platform that was expandable and adaptable to be used as a digital health solution platform, which arose from the need to manage cases in the context of syphilis [29] and integrate surveillance and health care actions. Its main objectives are to carry out control, integration, monitoring, and transparency in epidemiological surveillance actions and primary health care, focusing on case management. It is available at: https://salus2.lais.ufrn.br/.

The Salus Platform was developed in 2021 to manage syphilis cases through a set of requirements previously analyzed by the LAIS/UFRN research team. Salus emerged in the scenario of the COVID-19 pandemic. Despite being designed to manage syphilis cases, it has already proven effective at monitoring COVID-19 indicators [29], which allowed it to be tested in the context of disease case management. In addition, this was an opportunity to improve several algorithms and intelligent computational methods, as there was a large volume of data available, which would not be possible in the context of syphilis since, in the municipality where it was initially implemented (Natal/RN), the number of cases was relatively more minor [35]. The system is based on MoH PCDTs, which guide procedures for testing, monitoring, and treating infections. However, Salus is a digital tool that has several essential functions in the context of syphilis. One is real-time case reporting, which is the heart of the system.

The other activities branch off from this, and case management strategies emerge. For example, before the implementation of Salus, the process of notifying cases was precariously performed with manual data entry into spreadsheets, which generated a process that took an average of 2 days to complete [35]. On the other hand, after implementing Salus, notifications are handled in less than 1 h [35]. Another area that drives Salus is “Case Management” (some of these screens are shown in Appendix A Figure A1, Figure A2, Figure A3, Figure A4, Figure A5, Figure A6, Figure A7, Figure A8 and Figure A9). It is possible to view the patient’s entire history (exams, treatment, conduct) from the first entry into the care unit and follow each case until its outcome. In addition, it is possible to list some facilities and functionalities of the Intelligent Disease Monitoring System in Primary Care and Epidemiological Surveillance, as seen in Appendix B.

The Salus Platform remains under development by the management team and should present new features. However, as expected developments, the following stand out: better integration with e-SUS AB; validation of the Integration Agent with SINAN; new indicators related to Syphilis for Health Surveillance; new indicators related to Syphilis for Primary Care; new Integrated Indicators for Primary Care and Surveillance; improvement of the platform for greater adherence to the requirements of the General Law for the Protection of Personal Data (LGPD); and improved integration with other ecosystem systems or even outside the ecosystem [35].

A strong Salus strategy is centered on the generation of patient data, their social situation, treatments, tests performed, and results of similar health diagnoses from the three levels of health care, such as transparency and follow-up of each patient by professionals involved in care spheres, such as health surveillance managers and agents. A platform that makes the correct data available, with more precision and clarity, that is adapted to implementation for a more significant number of people infected with its services, are factors that strengthen health strategies and ensure that essential data are available.

The platform is made up of “actors” (nurse, physician, sanitary personnel, pharmacist, dentist, specialized care coordination, system administrator, and municipal management) who participate, within specific areas, in the management, insertion, and monitoring of data so that all professionals interact in patient care and follow-up and visualize the necessary data. Despite the pioneering idea of the system being the management of syphilis cases, the platform has the total capacity and instruments to accompany the management of cases of other notifiable diseases, for example, AIDS, Tuberculosis, and Viral Hepatitis [36].

Salus is a tool that integrates with the systems of the National Health Data Network (RDNS), National Card of the Unified Health System (CNS), National Register of Health Establishments (CNES), Information System of the National Immunization Program (SIPNI), and Exams through the Environmental Manager Laboratory (GAL), in addition to interconnection with the Brazilian Institute of Geography and Statistics (IBGE), Ministry of Health, and Municipal Health Secretariats [35], making a transparent and interconnected virtual environment with federal government systems and related bodies (Figure 1—Overview of the architecture of the Salus Platform).

The system has interconnected facilities and functionalities to make the platform accessible, transparent, fast, portable, and secure. They are tools that assist in the demand for data entry and the daily monitoring of case management, in addition to assisting in the process of notification and decision-making by managers of the primary health unit. The Salus Platform operates on the Hyper Text Transfer Protocol Secure (HTTPS) data transmission protocol, which ensures that all information is transmitted securely in the client–server architectural model, including the patient data. In addition to this security issue imposed at the application layer in the TCP/IP protocol stack, the Salus Platform uses login and password mechanisms for personalized access. Therefore, each user has a unique identification in the system and can only access it via password. Furthermore, Salus uses a log model that stores all actions performed in the system, so it is possible to track who performed specific procedures on it. Notably, this platform was implemented with a robust firewall structure that imposes levels of information security, such as monitoring and control against intrusions. In this context, all technological artifacts that make up its architecture are periodically updated to avoid security breaches.

Due to its functionalities, the Salus Platform is being implemented in several Brazilian municipalities. Training is carried out with members of the surveillance, primary care, and specialized care teams directly in the “test module” to adapt them to screens and resources, recording hypothetical cases as an active methodology. The professional learns using a “mirror” of the official version. Figure 2 shows which Brazilian cities/states Salus was implemented in until the time of submission of this study, in addition to pointing out the number of people who benefited from the implementation of the platform.

### 1.3. Data Entry Tools in Health Care in Brazil

The HIS in Brazil is part of the MoH Health Information and Technologies control and governance policies, which could counteract the fragmenting tendency of these systems. However, they could be better inducers of governments in the search for integrated solutions. This results in limited effectiveness, causing the system fragmentation scenario to persist [18,37].

In Brazil, the lack of systems that manage syphilis cases and the inefficiency of the few existing ones that offer to manage the data is evident. They are fragmented systems, without data interconnection and with duplicate information, difficult access, and delay in transmitting content, thus making it difficult to improve the quality of health services for the population, especially in the context of syphilis. In addition, these systems have differentiated proposals, which project an expectation of management of syphilis cases in Brazil. However, they do not have the fundamental characteristics that a management system needs, in addition to presenting inconsistencies in the information [15,16,17,19,38,39].

In this study, we surveyed the functionality of some systems used for cases of diseases in Brazil; however, we verified beforehand that they need to articulate with other systems, and there needs to be more information about delays in capturing data. A lack of these factors generates a delay in notification and makes it difficult to make the necessary decision in conducting the case.

Given this, developing a health information system that manages syphilis cases to end the noise in information, avoids unnecessary hospitalizations, and helps in the effectiveness of diagnosis and treatment. Adequate registration in information systems for syphilis, with reliable information from mandatory notification files and maintenance of the quality of comprehensive patient care, is necessary to reduce the incidence of infection to the levels targeted by the WHO. Poor record-keeping can compromise the organization and planning of care [40].

## 2. Materials and Methods

A descriptive, exploratory study was developed between February and October 2022 at three different times. Initially, a search was carried out in selected databases with comprehensive coverage of publications in the area of health (PubMed, Scopus, and Web of Science), which dealt with national health information systems. Then, as a selection criterion, only health information systems in Brazil that could present functional similarities with Salus were analyzed using the following descriptors: Health Information Systems, Health Informatics, and Public Health Informatics. In the second stage, information about each system, its functionalities, and proposals was sought. Finally, the third stage consisted of analyzing and crossing each system’s features with Salus’s functions to assess whether Salus fully meets some areas and can be indicated as a tool for improving health management in the context of syphilis. The objective was to assess whether the main tools available and used in data management in the Brazilian health service are sufficiently qualified to meet the demands of syphilis and other diseases compared to the tool under study (Salus).

Considering this context, this study aims to answer the following research problem: How do the Health Information Systems offered in Brazil contribute to managing cases of compulsorily notifiable diseases?

The Information Systems surveyed in this study were AMAQ (Self-Assessment to Improve Access and Quality of Primary Care—evaluation of the primary health care work process), CADSUS (User Registration System), CNES (National Register of Health Establishments—register of health establishments), e-SUS AB/SISAB (Health Information System for Primary Care—support for care management and control and monitoring of activities and procedures performed in primary health care), GAL (Laboratory Environment Management System—control and monitoring of laboratory results of diseases and conditions of public health interest; support the management of state public health laboratories), PMAQ-AB (National Program for Improving Access and Quality of Primary Care—control, monitoring, and evaluation of APS programmatic actions and work processes), SARGSUS (Support System for the Preparation of the Management Report—support to municipal management for the preparation and submission of the Annual Mana Gement Report (RAG) to the Health Council), SIA (Outpatient Information System—control and monitoring of the performance of outpatient procedures), SIAB (Primary Care Information System—control and monitoring of activities and procedures performed in Primary Health Care), SIASI (Indigenous Health Information System—control and monitoring of demographic information and health care for indigenous peoples), and SIS PRENATAL (Information System for Monitoring and Evaluation of Prenatal, Childbirth, Puerperium and Children—control and monitoring of health care for pregnant women, postpartum women, and newborns).

In this last stage, the implementation of Salus (from July 2021) in several Brazilian municipalities was commenced. By the time this article was submitted, more than one thousand Brazilian municipalities had already implemented Salus in their health services. In future studies, satisfaction questionnaires will be applied among users of the system, and data analysis will be carried out to evaluate the management of syphilis cases after the implementation of the Salus Platform.

## 3. Results

Numerous examples of systems that propose to collect data were revealed through searches in the literature for information systems that manage cases. However, they are far from proposing to manage cases, mainly in syphilis.

The comparative analysis of the Salus Platform with the other HIS tools showed that this system is an intelligent public health management tool that performs monitoring, surveillance, and management of cases of compulsory notification in primary and specialized care, fully integrated with epidemiological surveillance actions of SUS in Brazil. However, this analysis demonstrated systems with the most diverse functions, but they were fragmented, with communication problems, data transmission issues, and lack of interoperability between systems [17,18,19], generating fragmented data and discontinuity in care.

Furthermore, a lack of platforms that integrate cases between primary and specialized care, epidemiological surveillance and management, and sectors of extreme relevance in fighting, controlling, preventing, and monitoring disease cases was observed. This evidence points to some gaps that delay the outcome of demands and make routine therapeutic follow-up difficult, in addition to interfering with the vital purpose of follow-up in the epidemiological surveillance of diseases. Table 1 demonstrates this comparison and how the Salus Platform addresses these deficits, fully meeting its functionalities concerning managing and supervising cases in Brazilian public health. It should be remembered that this study mentioned only a small number of the systems in use in health care in Brazil. The purpose is to demonstrate how many functioning systems can be easily replaced by only one with all the necessary functionalities to manage cases, assist in decision-making, follow-up with the patient, notify the necessary cases, and reduce public expenses with hospitalizations that are often mistaken.

According to Table 1, the CADSUS system serves only to register SUS users. Registration of health establishments requires another system, the CNES. The e-SUS AB/SISAB supports the management of care and control and monitoring of activities and procedures performed in primary health care. However, it does not establish an interface between correlated diseases, in addition to not alerting the results of tests not performed, alterations, injectable drug applications, and abandonment alerts. The GAL system serves only to control and monitor the laboratory results of diseases and conditions of interest in public health but does not trigger alerts of cases of essential diseases for monitoring or diagnosis, in addition to incomplete data. It does not receive data or send electronic data. There is no interchangeability. In addition, two other systems are needed to control and monitor the activities and procedures carried out in Primary Health Care (SIAB) and to control and monitor the performance of outpatient procedures (SAI), without which, it is not possible to receive transfers of SUS financial resources to maintain the activities of health establishments and serve the population. Finally, if addressing the specific needs and monitoring demographic information and health care for indigenous peoples (SIASI) or controlling and monitoring health care for pregnant women, puerperal women, and newborns (SIS PRENATAL), two other independent systems that also require data entry separately will be necessary, contributing to an enormous rework and risk of recording inaccurate and incomplete data due to the human limitations of the system operators.

Salus can fully meet the needs related to citizen transparency. The pillars of the system are presented, including the epidemiological bulletin of syphilis cases by municipality and the privacy policy, as well as the validator of documents issued by Salus; it is also integrated into a public setting. The Indicator Panel includes cases of syphilis by type (congenital, pregnant, and acquired); records of performed exams; diagnosed cases; handled cases; patients with incomplete treatment or discharge by VDRL.

With Case Management, you can receive a list by using filters, in addition to being able to export data. It is also possible to initiate cases of CS, acquired syphilis or MS, and exposed children, update and close the therapeutic plan, record doses, record follow-up exams, consult case histories, register contacts, register occurrences, register permissions, monitor appointments, update user data, generate a SINAN form in PDF, generate a Salus form in PDF, and generate a receipt.

The Alerts and Notifications Center allows you to filter alerts by reason, neighborhood, and date, among others. For example, it can automatically trigger alerts for acquired syphilis, alerts for congenital syphilis, alerts for syphilis in pregnant women, alerts for exposed children, and late case alerts, in addition to being able to search for alerts by name, Individual Registration (CPF), or CNS.

As for the management of notifications posted on SINAN, it allows for confirmation of the entry made on SINAN, detailing data entry on SINAN, and filtering notifications by type, date, and process manager, among others. Furthermore, searching for unconfirmed SINAN notifications by name, CPF, CNS, or RG is also possible.

Management Reports show quantitative data on diagnosed cases, congenital syphilis, acquired syphilis, syphilis in pregnant women, exposed children, cases of abortion due to syphilis, cases of stillbirth due to syphilis, print queue management, email notifications, PDF data export, data export in spreadsheets, custom filters, individual notification, clinical and epidemiological data, treatment, conclusion/outcome of the case, and investigator/case manager.

There is the Interactive Map with Georeferencing of Cases (GEOSalus) that allows the objective visualization of syphilis cases by neighborhood, diagnoses by community, in progress by area, closed by neighborhood, and lost by the establishment, with active search, visualization of unlimited data and data by neighborhood, data export in PDF and data export in spreadsheets. The interactive map with Georeferencing of Cases (GEOSalus) allows the objective visualization of syphilis cases by neighborhood, diagnoses by community, in progress by neighborhood, closed by the community, and lost by the community, with active search, visualization of unlimited data and data by neighborhood, data export in PDF, and data export in a spreadsheet. Finally, it is possible to gain quick access to epidemiological bulletins, password recovery, privacy policy, and document validator.

In general terms, the essential functionalities presented by the Salus Platform led Brazilian municipalities to request that it be customized to assist in managing cases of other diseases.

In this study, the functionalities of Salus were demonstrated and compared to current Brazilian systems. As a result, Salus demonstrates functionalities that surpass all others and fulfills the demands of case management superiorly to the systems currently used in the country.

## 4. Discussion

Surveys and visits carried out by researchers from the Syphilis No! Project to health units in a large part of Brazil point to problems in the management of syphilis cases, mainly related to the lack of integration of data from surveillance and primary health care, the lack of communication between surveillance systems, and the inconsistency of data in the notification process [22,30]. Analogously, even in the context of syphilis, erroneous information and duplicate notifications occur from professionals who, on the recommendation of the Clinical Protocols and Therapeutic Guidelines (PCDT) of the MoH, report cases of children suspected or exposed to syphilis as being congenital syphilis, without the proper epidemiological investigation to conclude the case. This conduct leads to severe problems affecting the child, the family, and society. These are children who will be denied hospital discharge, will be hospitalized for ten days in an intensive care unit (ICU), accompanied by different professionals (nurses, physiotherapists, neurologists, speech therapists), subjected to treatments, which in most cases would only need be followed up to 2 years of age [41,42].

The comparative analysis of the Salus Platform with the other HIS tools showed that this system is an intelligent public health management tool that performs monitoring, surveillance, and management of cases of compulsory notification in primary health care, fully integrated with the SUS epidemiological surveillance actions in Brazil. Furthermore, despite having been developed for the response to syphilis with a solid architecture, Salus is adaptable and can be moulded according to the needs of each municipality so that its manager can integrate cases of other notifiable diseases [35]. Since its implementation, this platform has proven to be robust and secure as a data control and management tool. Through the platform, it is possible to mediate and better manage cases of syphilis in the municipality in order also to help eliminate vertical transmission of the disease.

For better management of cases of people with syphilis and data monitoring by technicians and servers working in this process, the platform works integrated with a center for monitoring cases and alerts triggered by the platform [35]. It is essential to highlight that in almost all Brazilian municipalities, the centers for compulsorily notifiable diseases or epidemiological surveillance still work heavily with data compiled in spreadsheets, which leads to significant delays in the preparation of bulletins and epidemiological reports. The delay in providing information for decision-making or data being recorded manually and wrongly can easily lead a manager to make late or wrong decisions. The Salus Platform is available to all municipalities that wish to change this reality. Likewise, it may be useful in cases of leprosy and tuberculosis, which often appear together with syphilis acquired in groups with different vulnerabilities, such as the homeless population; sex workers [43]; shelter residents; indigenous [44,45]; users of alcohol and other drugs; injecting drug users; persons deprived of liberty [46]; people in situations of sexual violence; history of other STIs [47]; multiple sexual partnerships; residing in an area without Primary Care coverage; history of mental health problems; and individuals not covered by the family health strategy.

In addition to all the functionalities, with the “Salus Prenatal” module and all these records, it will be possible to make available a “pregnant woman’s digital notebook”. Cases of illegible registration, incomplete mandatory fields, card loss, and miscellaneous incompleteness will be minimized or non-existent. With this, when the pregnant woman arrives at a maternity hospital, for example, the health team will be able to consult the prenatal care team, understand if the case has adequate prenatal care, and establish the appropriate measures for the puerperium.

Specifically, as an advance in the gestational context, the “Salus Prenatal” module (still to be implemented) proposes to carry out the control, integration, follow-up, and surveillance of assistance to pregnant women, childbirth, and the puerperium, aiming at humanization of assistance and strengthening of the strategy to face the transmission of diseases, including syphilis.

It is possible to register individual comorbidities such as identifying arterial hypertension, heart diseases, diabetes, chronic kidney disease, anemia, blood transfusions, neuropsychiatric diseases, viruses (rubella and herpes), surgery (type and date), allergies, leprosy, and tuberculosis. In addition to the pregnant woman, the partner is monitored to generate alerts to carry out the active search regarding when he needs the next consultation or treatment or even screening through exams. It is possible to monitor the performance and results of records of the most requested exams in each trimester of pregnancy.

First-trimester exams include blood typing and Rh factor; indirect Coombs, if the pregnant woman is Rh negative; fasting glucose (also called diabetes test); TSH and free T4 dosage; serology for syphilis, rubella, HIV, hepatitis B and C, IgM, and toxoplasmosis IgG and cytomegalovirus (only for pregnant women in the risk group); urine culture and type I urine; initial obstetric ultrasound, to confirm the date of pregnancy, as well as whether the embryo is developing correctly and whether the pregnancy is single or multiple; morphological ultrasound, to assess whether there is a risk of the fetus having a chromosomal syndrome, such as Down syndrome. Second-trimester exams include oral glucose tolerance test to detect gestational diabetes; morphological ultrasound, in which it is possible to identify fetal malformations, but also to see the sex of the baby. Finally, third-trimester exams include blood count; serology for some diseases, such as syphilis, HIV/AIDS, hepatitis B and C, and toxoplasmosis; screening for group B streptococcus; and obstetric ultrasound to assess fetal growth [42].

The data obtained in this work can support professionals and health managers in managing syphilis cases in Brazil.

## 5. Conclusions

The study compares the diversity of systems available in Brazilian public health, pointing out the absence of systems that manage cases in the context of syphilis in Brazil. Unfortunately, many systems in use have yet to achieve the purpose for which they were developed. Instead, they lead to fragmented data, diagnostic delays, incomplete case management, and data loss. These factors result in high and unnecessary public spending, delays in emergency care, high rates of illness, inconsistent reporting, and inadequate treatment. In this scenario, the opportunity arose to develop a platform to fill this worrying public health management gap, allowing for monitoring, follow-up, and case management, and, crucially, interconnecting surveillance and primary health care systems. It is a fact that the process of notification and management of syphilis cases in Brazil requires significant improvements and investments.

The study also contemplates other future opportunities for improving and adapting the Salus Platform, aiming to expand its functionalities to serve the entire population in several areas of health and, thus, to become the first consolidated Brazilian system as a digital solution tool in health management, linking surveillance and primary health care.

## Figures and Tables

**Figure 1 ijerph-20-05258-f001:**
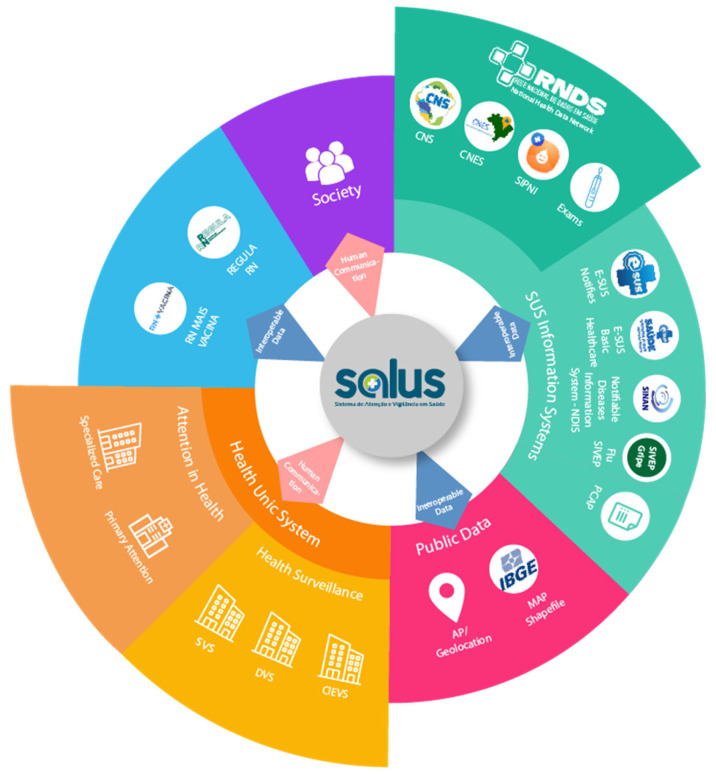
Overview of the architecture of the Salus Platform. Source: adapted from [35].

**Figure 2 ijerph-20-05258-f002:**
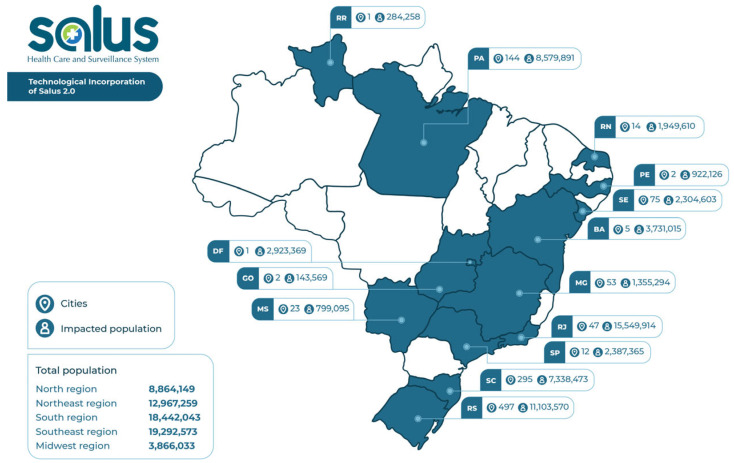
Implementation map of the Salus Platform in the Brazilian states. Source: elaboration of the authors.

**Table 1 ijerph-20-05258-t001:** Comparative analysis of the functionalities of the Health Information Systems (SIS) used in Brazil between 2010 and 2018, with those contemplated in Salus. Source: Adapted from [18].

Other Systems	Functionalities	System Salus Syphilis	Comparative Analysis
AMAQ—Self-Assessment to Improve Access and Quality of Primary Care—Evaluation of the primary health care work process	Evaluation of the primary health care work process	Indicators panel; Case Panel; Management Reports; Management of notifications launched on SINAN; Alerts and Notifications Center.	Fully meets
CADSUS—SUS User Registration System	Register of SUS users	Case Management; Active Search; Indicators panel; Case Panel; Management Reports.	Fully meets
CNES—National Register of Health Establishments—Register of health establishments	Register of health establishments	Link Management	Fully meets
e-SUS AB/SISAB—Health Information System for Primary Care—Support for care management and control and monitoring of activities and procedures performed in primary health care	Support for care management and control and monitoring of activities and procedures performed in primary health care	Indicators panel; Case Panel; Management Reports; Management of notifications launched on SINAN; Alerts and Notifications Center; Case Management; Active Search.	Fully meets
GAL—Laboratory Environment Management System—Control and monitoring of laboratory results of diseases and conditions of public health interest; supports the management of state public health laboratories.	Control and monitor laboratory results of diseases and conditions of public health interest; supports the management of state public health laboratories.	Alerts and Notifications Center; Case Management.	Fully meets
PMAQ-AB—National Program for Improving Access and Quality of Primary Care—Control, monitoring, and evaluation of APS programmatic actions and work processes	Control, monitor, and evaluation of APS programmatic actions and work processes	Indicators panel; Case Panel; Management Reports; Management of notifications launched on SINAN; Alerts and Notifications Center; Case Management; Active Search; Register User; Link Management; Interactive Map with Georeferencing of Cases (GEOSalus); Citizen Transparency.	Fully meets
SARGSUS—Support System for the Preparation of the Management Report—Support to municipal management for the preparation and submission of the Annual Management Report (RAG) to the Health Council	Support to municipal management for the preparation and submission of the Annual Management Report (RAG) to the Health Council	Indicators panel; Case Panel; Management Reports; Management of notifications launched on SINAN; Alerts and Notifications Center; Interactive Map with Georeferencing of Cases (GEOSalus); Citizen Transparency.	Fully meets
SIA—Outpatient Information System—Control and monitoring of the performance of outpatient procedures	Control and monitoring of the performance of outpatient procedures	Case Management; Active Search; Indicators panel; Case Panel; Management Reports; Interactive Map with Georeferencing of Cases (GEOSalus).	Fully meets
SIAB—Primary Care Information System—Control and monitoring of activities and procedures performed in Primary Health Care	Control and monitoring of activities and procedures performed in Primary Health Care	Case Management; Active Search; Indicators panel; Case Panel; Management Reports; Interactive Map with Georeferencing of Cases (GEOSalus).	Fully meets
SIASI—Indigenous Health Information System—Control and monitoring of demographic information and health care for indigenous peoples	Control and monitoring of demographic information and health care for indigenous peoples	Case Management; Active Search; Indicators panel; Case Panel; Management Reports; Interactive Map with Georeferencing of Cases (GEOSalus).	Fully meets
SIS PRÉ-NATAL—Information System for Monitoring and Evaluation of Prenatal, Childbirth, Puerperium and Children—Control and monitoring of health care for pregnant women, postpartum women, and newborns	Control and monitoring of health care for pregnant women, postpartum women, and newborns	Case Management; Active Search; Indicators panel; Case Panel; Management Reports; Interactive Map with Georeferencing of Cases (GEOSalus).	Fully meets

## Data Availability

Not applicable.

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
