# Peer review of "Salus Platform: A Digital Health Solution Tool for Managing Syphilis Cases in Brazil—A Comparative Analysis"

_ijerph, 2023, doi:10.3390/ijerph20075258_

Round 1
Reviewer 1 Report
This manuscript introduces a digital health solution tool for managing 2 syphilis cases in Brazil, which is of great significance for the digital and modern management of syphilis and other infectious diseases. The abstract of this paper should be more specific and should be supplemented with a brief description of the research methods and objects. The results section of the summary should complement the specific functional modules of the platform, not simply list gaps with existing systems. The content of platform data security and privacy handling must be added.Author Response
March 11, 2023.
Dear Reviewer,
We appreciate the suggestions given to our manuscript by the reviewers, who brought the message across clearly and helped keep the focus on essential aspects of this study. The suggestions were contemplated, and the questions were answered. Corrections were made in English, insertion of other references relevant to the work and some paragraphs were rewritten for better understanding by the reader.
Reviewer 1 | 28 Feb 2023 | 15:59:51
This manuscript introduces a digital health solution tool for managing 2 syphilis cases in Brazil, which is of great significance for the digital and modern management of syphilis and other infectious diseases.
- The abstract of this paper should be more specific and should be supplemented with a brief description of the research methods and objects.
Response: Done. We rephrased the sentence (Lines 30-79).
- The results section of the summary should complement the specific functional modules of the platform, not simply list gaps with existing systems.
Response: Done. We rephrased the sentence (Lines 36-42).
- The content of platform data security and privacy handling must be added.
Response: Done. We rephrased the sentence (Lines 518-533).
Best regards,
Talita Brito

Reviewer 2 Report
Introduction seems to be lengthy. Since the manuscript was communicated as Research article, so much information is not required. Also Syphilis No! Project was already discussed on the Author’s previous article. Hence revise the manuscript to focus on comparative analysis of what is exist and what is required?
Authors have described that the Salus can meet the requirement of the health information system description. However, the objective of the manuscript “Comparative analysis” was not done or not presented in the manuscript to describe the advantage of Salus.
This manuscript does not describe how many patients were enrolled and how the monitoring of the patient using Salus improved patient’s treatment outcome is not discussed.
Authors have mentioned qualitative analysis was done in the abstract. However, the manuscript is not discussing either qualitative and quantitative.
I do not understand the information provided in Appendix A.
Author Response
March 11, 2023.
Dear Reviewer,
We appreciate the suggestions given to our manuscript by the reviewers, who brought the message across clearly and helped keep the focus on essential aspects of this study. The suggestions were contemplated, and the questions were answered. Corrections were made in English, insertion of other references relevant to the work and some paragraphs were rewritten for better understanding by the reader.
Reviewer 2 | 01 Mar 2023 | 22:37:15
- Introduction seems to be lengthy. Since the manuscript was communicated as Research article, so much information is not required. Syphilis No! project was already discussed on the Author’s previous article. Hence revise the manuscript to focus on comparative analysis of what is exist and what is required?
Response: Done. We rephrased the sentence (Lines 82-573).
- Authors have described that the Salus can meet the requirement of the health information system description. However, the objective of the manuscript “Comparative analysis” was not done or not presented in the manuscript to describe the advantage of Salus.
Response: Done. We rephrased the sentence (Lines 372-381; 553-574; 583-592; 626-747; table 1; 767-794; 847-850; 920-930; 992-1002).
- This manuscript does not describe how many patients were enrolled and how the monitoring of the patient using Salus improved patient’s treatment outcome is not discussed.
Response: The discussion of treatment data is very important in implementing a case management tool; however, this data was not contemplated because it is different from the objective of this work. Regardless of, these data already exist, and their analyzes will be presented in future work.
- Authors have mentioned qualitative analysis was done in the abstract. However, the manuscript is not discussing either qualitative and quantitative.
Response: Done. We rephrased the sentence.
- I do not understand the information provided in Appendix A.
Response: These are some screens of the Salus System, focused on its mainly panels. They were translated into English.
Best regards,
Talita Brito

Reviewer 3 Report
This is a well written manuscript and I applaud the authots for their innovative approach to managing syphilis in Brazil. The use of information technology is completeley underutilized and efforts such as these should be considered in the US and globally.
It appears through this manuscript that this data/results will be entered by health ministry personnel and disease intervention specialists, or will it be entered by clinicians, or both? OR is it electronically transferred from the public or private labs? Will data be accessed by providers specific only to the providers and health officials? How is that process secure and private?
Our national and local health authorities do not disclose an information regading lab results/managment or past treatment unless we are providers for that patient. The patient also does not have access to her/his own results. This often poses a challenge for us to manage positive results given the fact that these offices close at 4:30 pm - so the providers have to wait.
I only ask for some clarity regarding the process of transfer of data, who is responsible and who manages the abnormals. I am intigued by this and hope that with a more detailed desription, we can determine feasibility in other settings.
The introduction can be shortened and methods as described above should be detailed a bit more. Also, for the tools and figures at the end of the manuscript, is it possible to also demonstrate these in English?
I am very excited about this work and appreciate the opportunity to review.
Author Response
March 11, 2023.
Dear Reviewers,
We appreciate the suggestions given to our manuscript by the reviewers, who brought the message across clearly and helped keep the focus on essential aspects of this study. The suggestions were contemplated, and the questions were answered. Corrections were made in English, insertion of other references relevant to the work and some paragraphs were rewritten for better understanding by the reader.
Reviewer 3 | 22 Feb 2023 | 13:23:38
This is a well written manuscript and I applaud the authors for their innovative approach to managing syphilis in Brazil. The use of information technology is completeley underutilized and efforts such as these should be considered in the US and globally.
- It appears through this manuscript that this data/results will be entered by health ministry personnel and disease intervention specialists, or will it be entered by clinicians, or both? OR is it electronically transferred from the public or private labs?
Response: Done. (Lines 398-402). By Ordinance No - 204 of February 17, 2016. VI - Compulsory notification: mandatory communication to the health authority, carried out by doctors, health professionals or those responsible for health establishments, public or private, about the occurrence of suspected or confirmed illness, injury or public health event, described in the annex, which may be immediate or weekly;
- Will data be accessed by providers specific only to the providers and health officials?
Response: All data available on the Salus platform are only accessed by health professionals, who are obligatorily accredited and authorized by the health authorities. Incorporating the Salus platform into the Unified Health System (UHS) in Brazil to integrate Primary Health Care into Health Surveillance occurs through follow-up and monitoring by the Health Surveillance Secretariat (HSS) of the Ministry of Health of Brazil. It should also be noted that, in this process of incorporation into the SUS, Salus also implements the modelling of care networks, ranging from pregnant women (prenatal care) to children exposed to syphilis.
- How is that process secure and private?
Response: No private data is public on Salus, so only duly accredited health authorities can access patient data, for example. Concerning public areas, the system only presents aggregated data and publicly accessible information, which comply with Law No. 12,527, of November 18, 2011 (Regulates access to information provided in item XXXIII of art. 5, in item II of § 3 of art. 37 and in § 2 of art. 216 of the Federal Constitution; amends Law No. 8,112, of December 11, 1990; revoke Law No. 11,111, of May 5, 2005, and provisions of Law No. 8,159, of January 8, 1991; and takes other measures). In addition, Salus follows Law nº 13.709/2018 - GLPD (General Law for the Protection of Personal Data).
Our national and local health authorities do not disclose information regarding lab results/management or past treatment unless we are providers for that patient. The patient also does not have access to her/his own results. This often poses a challenge for us to manage positive results given the fact that these offices close at 4:30 pm - so the providers have to wait.
- I only ask for some clarity regarding the process of transfer of data, who is responsible and who manages the abnormal. I am intrigued by this and hope that with a more detailed description, we can determine feasibility in other settings.
Response: The Salus platform performs all data transfer over the HTTPS protocol, always based on the client-server model. Therefore, data is sent/transferred online. Health professionals perform data curation/management during data entry; however, the system has filters and locks that prevent abnormal data from being included. It is essential to highlight that the Salus Platform has already been running for two years and that system users validate this process at operational and managerial levels. Integrating data with the legacy systems of the Brazilian Ministry of Health occurs through the National Health Data Network (RNDS); therefore, it follows the normative criteria that require approval of the data transfer process by Greetings. Therefore, all transfers between Salus and the other Brazilian Ministry of Health platforms are approved (Lines 518-533).
- The introduction can be shortened and methods as described above should be detailed a bit more.
Response: Done. We rephrased the sentence (Lines 82-573; 577-619).
- Also, for the tools and figures at the end of the manuscript, is it possible to also demonstrate these in English?
Response: Thanks for the observation. The figures were translated into English.
I am very excited about this work and appreciate the opportunity to review.
Best regards,
Talita Brito

Round 2
Reviewer 2 Report
Check with Editors